# Valorization of *Quercus suber* L. Bark as a Source of Phytochemicals with Antimicrobial Activity against Apple Tree Diseases

**DOI:** 10.3390/plants11243415

**Published:** 2022-12-07

**Authors:** Eva Sánchez-Hernández, Vicente González-García, José Casanova-Gascón, Juan J. Barriuso-Vargas, Joaquín Balduque-Gil, Belén Lorenzo-Vidal, Jesús Martín-Gil, Pablo Martín-Ramos

**Affiliations:** 1Department of Agricultural and Forestry Engineering, ETSIIAA, Universidad de Valladolid, Avenida de Madrid 44, 34004 Palencia, Spain; 2Department of Agricultural, Forestry and Environmental Systems, Agrifood Research and Technology Centre of Aragón, Instituto Agroalimentario de Aragón—IA2 (CITA-Universidad de Zaragoza), Avda. Montañana 930, 50059 Zaragoza, Spain; 3Instituto Universitario de Investigación en Ciencias Ambientales de Aragón, EPS, Universidad de Zaragoza, Carretera de Cuarte s/n, 22071 Huesca, Spain; 4AgriFood Institute of Aragon (IA2), CITA-Universidad de Zaragoza, Avda. Montañana 930, 50059 Zaragoza, Spain; 5Servicio de Microbiología, Hospital Universitario Rio Hortega, Calle Dulzaina 2, 47012 Valladolid, Spain

**Keywords:** antimicrobial, antifungal, brown rot, collar and root rot, cork oak, dieback, fire blight

## Abstract

Cork, an anatomic adaptation of the bark of *Quercus suber* L. through its suberization process, finds its main application in the production of bottle stoppers. Its processing results in a large waste stream of cork fragments, granulates, and dust, which may be susceptible to valorization. The work presented here explored the use of its extracts to inhibit the growth of phytopathogenic microorganisms associated with apple tree diseases. The in vitro antimicrobial activity of cork aqueous ammonia extract was assayed against four fungi, viz. *Monilinia fructigena* and *M. laxa* (brown rot), *Neofussicoccum parvum* (dieback), and *Phytophthora cactorum* (collar and root rot), and two bacteria, viz. *Erwinia amylovora* and *Pseudomonas syringae* pv. *syringae*, either alone or in combination with chitosan oligomers (COS). Effective concentration values of EC_90_ in the 675–3450 μg·mL^−1^ range, depending on the fungal pathogen, were obtained in growth inhibition tests, which were substantially improved for the conjugate complexes (340–801 μg·mL^−1^) as a result of strong synergism with COS. Similar enhanced behavior was also observed in antibacterial activity assays, with MIC values of 375 and 750 μg·mL^−1^ for the conjugate complexes against *P. syringae* pv. *syringae* and *E. amylovora*, respectively. This in vitro inhibitory activity was substantially higher than those exhibited by azoxystrobin and fosetyl-Al, which were tested for comparison purposes, and stood out among those reported for other natural compounds in the literature. The observed antimicrobial activity may be mainly attributed to the presence of glycerin and vanillic acid, identified by gas chromatography–mass spectroscopy. In the first step towards in-field application, the COS–*Q. suber* bark extract conjugate complex was further tested ex situ against *P. cactorum* on artificially inoculated excised stems of the ‘Garnem’ almond rootstock, achieving high protection at a dose of 3750 μg·mL^−1^. These results suggest that cork industrial leftovers may, thus, be a promising source of bioactive compounds for integrated pest management.

## 1. Introduction

The cork oak (*Quercus suber* L.) is a slow-growing, evergreen tree indigenous to the Mediterranean region [1]. The cork shields stem shoots and buds from outside threats [2]. Its qualities (lightweight, waterproofness, compressibility, superior acoustic insulation, low thermal conductivity, high energy absorption capacity, and resistance to friction, fire, and impact [3]) make it suitable for flooring and insulation applications, and it is also used as natural cork stoppers for bottles [4]. In cork processing, cork fragments, granulates and dust represent a large waste stream [5], which is susceptible to valorization.

The chemical composition of cork is influenced by its geographic origin, temperature and soil conditions, genetic factors, tree size and age, and cork harvesting regime [6]. The polymeric matrix of cork is mainly composed of suberin, lignin, and polysaccharides [7]. Coquet et al. [8] showed that extraction under acid conditions led to diacids (tetracosanedioic, docosanedioic, and eicosanedioic acids) and monoacids (palmitic, stearic, thapsic, tetracosanoic, docosanoic, and eicosanoic acids), while extraction under neutral conditions led to 2,6-heptanediol and long chain alcohols (tetracosanol, docosanol, eicosanol), as well as sterols and triterpens (*β*-sitosterol, stigmasterol, 3-friedelanol, friedelin, and betulin). Analyses of extractable phenolic compounds have also shown the presence of gallotannins, ellagitannins, dehydrated tergallic-C-glucosides or ellagic acid derivatives, and mongolicain [9]. Other extractives reported in the literature are *trans-*squalene, camphene, trans-3-pinanone, 1-terpinen-4-ol, vescalagin, castalagin, pyrogallol, glucosan, sitost-4-en-3-one, o-cymene, and quinic acid [10].

Recent studies report that cork oak has high antimicrobial activity; Borrero et al. [11] showed that cork compost inhibited the growth of *Fusarium* spp., *Pythium aphanidermatum* (Edson) Fitzpatrick, * Rhizoctonia solani* Kühn, and *Botrytis cinerea* Pers. A suberin film extracted from cork showed bactericidal action against *Staphylococcus aureus* NCTC8325 and *Escherichia coli* (Migula) Castellani & Chalmers [12]. A methanolic leaf and stem extract exhibited inhibitory activity against *Bacillus subtilis* (Ehrenberg) Cohn, *Streptococcus pneumoniae* (Klein) Chester, *E. coli,* and *S. aureus*, and against *Aspergillus niger* Tiegh., *Penicillium* sp. and *Fusarium oxysporum* Schlecht. fungal strains that was reportedly better than those of other species of the genus *Quercus* [13]. 

Despite cork’s well-known and varied qualities, little attention has been paid to its potential use as a biorational against phytopathogens in fruticulture, in particular, to protect plants that belong to the *Rosaceae* family. Fungi of the genus *Monilinia*, especially *Monilinia fructigena* (Pers.) Honey and *Monilinia laxa* (Aderh. & Ruhland) Honey, cause brown rot disease in stone and pome fruits. In susceptible cultivars, these taxa spread to both young shoots and flowering buds, causing twig cankers and wilting of growing shoots, as well as fruit rot. *Monilinia laxa* causes significant losses of stone fruit in the field and after harvest [14,15]. *Neofussicoccum parvum* (Pennycook & Samuels) Crous, Slippers & A.J.L. Phillips causes fruit rot, cankers, and dieback [16]. As a result of climate change, it is becoming an emerging disease of *Rosaceae* plant species, which enhances the necessity for learning about its pathogenicity, particularly concerning apple varieties of significant economic value [17]. As for the oomycete *Phytophthora cactorum* (Lebert & Cohn) J. Schröt., it causes fruit rot, starting with wilt and eventually destroying the tissues, in apple, apricot, citrus, plum, and strawberry crops. Regarding bacterial diseases, fire blight, caused by *Erwinia amylovora* (Burrill) Winslow et al., is a serious global threat to the production of apples and pears [18], and the genus *Pseudomonas* is among the ten most widespread bacterial plant diseases worldwide. Specifically, *Pseudomonas syringae* pv. *syringae* van Hall causes bacterial leaf spot and cankers and can affect species from *Fabaceae*, *Cruciferae*, *Solanaceae*, and *Rosaceae* families [19].

Different cultural methods, chemical fungicides in the orchard, treatments on mummified fruits, and post-harvest storage at low temperatures can be used to manage diseases of apple trees [20]. However, regarding chemical fungicides, it should be noted that a gradual withdrawal of some substances is taking place, due to concerns about their detrimental effects on the environment and human health, the persistent threat of the emergence of resistance, and the emergence of new virulence alleles [21]. As an alternative, the European Union, through Regulation (EU) 2019/1009, Council Regulation (EC) 834/2007, Commission Regulation (EC) 889/2008, and Article 14 of Directive 2009/128/EC, encourages the use of formulations based on natural products in an integrated pest management context. 

In line with the latter approach, this work proposes the valorization of an aqueous ammonia extract of *Q. suber* bark as an antimicrobial agent for the protection of crops of the *Rosaceae* family, and, in particular, of *Malus domestica* Borkh. The main constituents were identified by gas chromatography–mass spectroscopy (GC–MS). The antimicrobial activity of the extract, both alone and as part of conjugated complexes with chitosan oligomers (COS), as well as the activity of its main bioactive compounds against the aforementioned phytopathogens, were tested in vitro. Finally, to test the preventive potential of the extract, an ex situ study was carried out on excised stems of a rootstock accession susceptible to *P. cactorum*.

## 2. Results

### 2.1. Identification of Phytochemicals by GC–MS

The GC–MS analysis of the aqueous ammonia extract of *Q. suber* bark revealed the presence of a hundred chemicals (Appendix A, Appendix A). Out of these, glycerin (7.7%), 4-hydroxy-3-[[1,3-dihydroxy-2-propoxy]methyl]-1H-pyrazole-5-carboxamide (4.4%), 2-azabicyclo[2.2.1]heptane (3.6%), 4-hydroxy-3-methoxy-benzoic acid (or vanillic acid) (3.3%), benzoic acid (2.7%), 4-hydroxy-benzoic acid (1.6%), azelaic acid (2.6%), 1-decene (2.3%), cyclopentadecane (2.2%), and α-amino-γ-butyrolactone (1.9%) were the main natural constituents, while N-hydroxycarbamic acid, 2-(propoxycarbonylamino) ethyl ester, and 3D-10D cyclosiloxanes were found to be contaminants, given that the former is a product from reactions that involve both monoethanolamine (MEA) and methyl diethanolamine (MDEA) degradation products [22] (and MEA is considered as a potential atmospheric pollutant, since it is a benchmark and widely utilized solvent in a leading CO_2_ capture technology [23]), and the latter originates from septum and column bleed [24,25]. The chemical structure of the former is presented in Figure 1.

### 2.2. In Vitro Antimicrobial Activity Assessment

#### 2.2.1. Antifungal and Anti-Oomycete Activity

The results of the antifungal/anti-oomycete susceptibility test are summarized in Figure 2. For all the assayed products, an increase in concentration led to a decrease in the radial growth of the mycelium, resulting in statistically significant differences. The aqueous ammonia extract of *Q. suber* showed antifungal activity comparable to that of COS, with minimum inhibitory concentrations of 1500 μg·mL^−1^ against taxa of the genus *Monilinia* and 750 μg·mL^−1^ against *P. cactorum*. However, the *Q. suber* extract did not inhibit mycelial growth in *N. parvum*, with a percentage inhibition of less than 50% for the highest concentration tested (1500 μg·mL^−1^). Concerning the main constituents of the extract, viz. glycerin, vanillic acid, and 2-azabicyclo[2.2.1]heptane (4-hydroxy-3-[[1,3-dihydroxy-2-propoxy]methyl]-1H-pyrazole-5-carboxamide could not be tested due to its unavailability from chemical suppliers), they presented better or similar inhibition values compared with those obtained by the extract. Specifically, glycerin and vanillic acid completely inhibited mycelial growth, with MICs ranging from 375 to 1500 μg·mL^−1^, whereas 2-azabicyclo[2.2.1]heptane was unable to fully inhibit the growth of any of the pathogens tested. The formation of conjugate complexes led to an improvement in terms of antifungal activity, as the COS–*Q. suber* extract resulted in complete inhibition at concentrations in the 375–1000 μg·mL^−1^ range, while full inhibition was observed at concentrations in the 250–1000 μg·mL^−1^ range for COS–glycerin and COS–vanillic acid (i.e., at doses lower than that observed for the COS–2-azabicyclo[2.2.1]heptane conjugate complex, with an MIC of 1500 μg·mL^−1^ against all pathogens). This improvement is more clearly observed in the effective concentration values summarized in Table 1.

To quantify the synergistic behavior observed for the conjugate complexes, synergy factors [13] were calculated according to the Wadley method (Table 2). Synergism (i.e., SFs > 1) was detected in all cases except for COS–2-azabicyclo[2.2.1]heptane against *P. cactorum*. 

For comparison and effectiveness purposes, the results of the experiments conducted with three conventional synthetic fungicides are presented in Table 3. The highest inhibition rates were recorded for a dithiocarbamate (mancozeb), demonstrating full inhibition of the mycelial growth of the four phytopathogens at one-tenth of the recommended dose (i.e., at 150 μg·mL^−1^). The organophosphorus fungicide (fosetyl-Al) fully inhibited the growth of *P. cactorum* at the recommended dose (i.e., 2000 μg·mL^−1^), but required a higher concentration to achieve full inhibition of the other three pathogens. Concerning the strobilurin fungicide (azoxystrobin), it was the least effective; it did not fully inhibit the growth of *M. laxa*, *M. fructigena*, or *N. parvum* at ten times the recommended dose (625 mg·mL^−1^), although it inhibited the growth of *P. cactorum* at 62.5 mg·mL^−1^. 

#### 2.2.2. In Vitro Antibacterial Assessment

Table 4 provides a summary of the antibacterial activity results against the two Gram-negative bacterial plant pathogens. In the case of the quarantine species *E. amylovora*, *Q. suber* bark extract was more effective than COS, with MIC values of 1000 and 1500 μg·mL^−1^, respectively. Two of its main constituents, glycerin, and vanillic acid, were more effective than the extract, with MIC values of 500 and 750 μg·mL^−1^, respectively. Regarding the activity against *P. syringae* pv. *syringae*, the aqueous ammonia extract, glycerin, and vanillic acid inhibited bacterial growth at 750 μg·mL^−1^, while a concentration of 1000 μg·mL^−1^ was required for COS conjugates. As for 2-azabicyclo[2.2.1]heptane, it did not prevent bacterial growth at the maximum dose assayed of 1500 μg·mL^−1^. 

As previously discussed for the antifungal/anti-oomycete activity, the formation of conjugated complexes with COS resulted in a synergistic effect, enhancing the antibacterial activity. In the case of *P. syringae* pv. *syringae*, the *Q. suber*–COS conjugate complex fully inhibited bacterial growth at a concentration of 375 μg·mL^−1^, although a higher concentration of 750 μg·mL^−1^ was required against *E. amylovora*. Regarding the conjugates of the extract constituents, COS–glycerin was the most effective against both bacteria, with MIC values of 375 μg·mL^−1^; followed by COS–vanillic acid, with MIC values of 500 μg·mL^−1^. Interestingly, the COS–azabicyclo conjugate resulted in full inhibition at 750 μg·mL^−1^, suggesting strong synergism, which may be tentatively ascribed to solubility enhancement.

### 2.3. Protection of Excised Stems against P. cactorum

Ex situ tests were conducted on excised stems from the ‘Garnem’ rootstock to assess the efficacy of the treatment against the phytopathogen for which the best in vitro results had been obtained, namely *P. cactorum* (Figure 3). At the lowest assayed dose, i.e., the MIC value obtained in the in vitro tests (375 μg·mL^−1^), no protection effect was observed, with canker lengths similar to those of the untreated stems (Table 5). At five times the MIC dose (1875 μg·mL^−1^), large cankers were also registered, but with significant differences compared with the controls. It was necessary to increase the dose up to 10 times the MIC (3750 μg·mL^−1^) to obtain high and statistically significant protection of the excised stems. 

## 3. Discussion

### 3.1. The Phytochemical Profiles

In general, the main compounds present in cork are terpenes, sterols, saccharides, suberin, lignin, and other phenolic compounds, although their composition may vary as a result of various factors, such as climate, region, age, or part of the tree [2]. To extract phenolic compounds from the bark, polar, organic solvents or hydromethanolic mixtures are mainly used, identifying essentially phenolic acids and aldehydes, coumarins, flavonoids, and tannins [9,26]. Recently, polyphenol recovery in *Q. suber* bark extracts was considerably increased by microwave-assisted extraction, using different proportions of water and alcohols, mainly obtaining *p*-coumaric, syringic, and sinapic acids [27]. In addition, a new method has been reported for the extraction of phenolic compounds from cork granulates using a mixture of water with propylene glycol [28]. However, to the best of the authors’ knowledge, this is the first time that an aqueous ammonia solution and ultrasonication have been used for the extraction of bioactive compounds from cork. Previously, our research group used this extraction method with *Quercus ilex* subsp. *ballota* (Desf.) Samp. [29] and *Uncaria tomentosa* L. barks [30]. 

The obtained hydroxybenzoic acid profile was different from those previously reported [8,10], but exhibited similarities with that of *Q. ilex* bark [29], i.e., the presence of hydroxyciclopentenones, methylimidazoles, and hexadecanoic and nonanedioic acid/esters. Concerning other phytochemicals, the pyrazole 4-hydroxy-3-[[1,3-dihydroxy-2-propoxy]methyl]-1H-pyrazole-5-carboxamide is analogous to pirazofurin (a C-glycosyl compound that is 4-hydroxy-1H-pyrazole-5-carboxamide in which the hydrogen at position 3 has been replaced by a *β*-D-ribofuranosyl group) with antineoplastic and antiviral properties [31]. 2-azabicyclo[2.2.1]heptanes are analogs of aliphatic monoamines (pyrrolidine; piperidine), distributed mainly in the *Piperaceae* and *Rutaceae* families, with described antifeedant, analgesic, antipyretic, anti-inflammatory and antioxidant activities [32]. Specifically, 2-azabicyclo[2.2.1]heptane was previously isolated from *Delphinium caeruleum* Jacquem. ex Cambess. [33]. The flavoring agent vanillic acid (or 4-hydroxy-3-methoxy-benzoic acid) is a phenolic acid associated with lignin following oxidation. It is the intermediate product in the two-step bioconversion of ferulic acid to vanillin [34]. Azelaic acid (or nonanodioic acid) is a naturally occurring acid found in grains such as barley, wheat, and rye, and it serves as a signal that induces the accumulation of salicylic acid, an important component of the defensive response of a plant [35]. Azelaic acid has been reported to manifest its antibacterial effects by inhibiting the synthesis of cellular proteins in both anaerobic bacteria (impeding glycolysis) and aerobic bacteria (inhibiting several oxidoreductive enzymes, including tyrosinase, mitochondrial enzymes of the respiratory chain, thioredoxin reductase, 5-α-reductase, and DNA polymerases) [36]. 1-decene was detected as one of many volatile organic compounds emitted from woodland vegetation, with an emission rate estimated to range from 0.5 to 5 µg·g^−1^ [37]. The phytochemical α-amino-γ-butyrolactone is a cleavage product of S-adenosylmethionine [38].

### 3.2. Mode of Action

To provide a tentative explanation of the observed antimicrobial activity, a discussion of the antimicrobial activity of each of the three main extract constituents that were tested in vitro is presented (although concurrent activity from other constituents of the extract and/or synergism among the various phytochemicals cannot be ruled out). 

Glycerin, a simple three-carbon tri-alcohol used as a carrier in many medicines and as a plasticizer in gelatin gel capsules, is known to be a bacteriostatic/fungistatic [39], which also features virucidal activity [40]. Concerning its applicability to the preventive and/or curative treatment of plants, glycerin as it is or in a water-based solution has been reported to possess fungicidal and bactericidal properties against some types of phytopathogen fungi and bacteria (*Alternaria alternata* (Fr.) Keissl., *Venturia inaequalis* (Cooke) G. Winter, *Plasmopara viticola* (Berk. & M.A. Curtis) Berl. & de Toni, *Cercospora beticola* Sacc., *Puccinia* spp., *Fusarium* spp., *Septoria* spp., *Botrytis* spp., *Taphrina* spp., *Phytophthora infestans* (Mont.) de Bary, *E. amylovora*, and *Erysiphe necator* Schwein.), although it was noted that it demonstrated greater fungicidal and bactericidal action when used in combination with other substances with a known antimicrobial action, allowing a reduction in the concentration of the latter [41].

The presence of glycerin in plant extracts has been reported, for instance, in *Plantago major* L. leaf extracts [42]; *Allamanda cathartica* L. [43]; *Cynodon dactylon* (L.) Pers. (with an activity comparable to that of streptomycin against *S. aureus*, *E. coli*, *Salmonella typhi* (Schroeter) Warren and Scott, *Proteus mirabilis* Hauser, and *S. pyogens*) [44]; *Salvadora persica* L. (with activity against *S. aureus* and *A. terreus*) [45]; and *Aphelandra squarrosa* Nees (with a glycerin content as high as 46% and strong activity against *E. coli*) [46].

Regarding vanillic acid, its presence has been reported in extracts with antimicrobial activity from other plants (Appendix A), although *Angelica sinensis* (Oliv.) Diels is currently the largest commercial natural source [47]. The mechanism of action of its antimicrobial activity against clinically important bacteria, by inducing cell lysis, damaging the cell membrane, and causing leakage of intracellular components, has been studied by Li et al. [48]; and it is considered a promising candidate antimicrobial agent not only to treat infections and as a surface disinfectant, but also as a food preservative in the food industry [49]. 

In relation to azabicyclo[2.2.1]heptane, the presence of azabicyclo derivatives in plant extracts has not yet been thoroughly studied. Nonetheless, 6-azabicyclo[3.2.1]octane was found to be present in an extract of *Azadirachta indica* A. Juss. and in *Ocimum sanctum* L. leaves [50]; 1-azabicyclo[3.1.0]hexane was reported in an extract of *Melia dubia* Cav. leaves [51]; and the highest concentration of an azabicyclo derivative to date was found in an extract of a *Dioscorea hispida* Dennst. tuber at 23.16% [52], all of which showed antimicrobial activity (Appendix A). Nonetheless, as noted above, in this work, the activity of azabicyclo[2.2.1]heptane was low when used alone, which may be tentatively ascribed to solubility problems, given that it improved upon conjugation with COS.

### 3.3. Antimicrobial Activity Comparison

#### 3.3.1. Comparison with Antimicrobial Activities Reported for Other *Q. suber* Extracts

The available studies on the antimicrobial activity of *Q. suber* extracts are summarized in Appendix A. Given that no data on activity against the same pathogens have been reported, the comparisons below should be taken with caution. Akroum [53] reported the high in vitro antifungal activity of an acetonic extract of *Q. suber* acorns against seven pathogenic fungi, with inhibition values in the 20–105 µg·mL^−1^ range (better than those reported here), which were attributed to the composition of the acorns, rich in phenolic acids and gallic acid derivatives, as well as in proanthocyanidins (present in different species of *Quercus* spp., mainly *Q. ilex* L., *Q. suber* L., and *Q. robur* L. [54]). However, the aforementioned values were obtained using extracts in acetone (70%), which calls into question their validity (due to the aggressiveness of the extraction medium used). Concerning methanolic or hydromethanolic extracts, Lahlimi-Alami et al. [55] tested the anticandidosic potential of a methanolic extract of cork against five different strains of *C. albicans*, with minimum inhibitory concentrations in the 12,500–50,000 µg·mL^−1^ range. These values were similar to those obtained by Hassikou et al. [56] who used methanolic extracts of cork and leaf. This may explain why Touati et al. [6] reported that an hydromethanolic extract of cork failed to inhibit bacterial growth of *Listeria innocua* (ex Seeliger and Schoofs 1979) Seeliger 1983 and *E. coli*, given that the maximum concentration assayed was noticeably lower (3000 µg·mL^−1^). In comparison, the aqueous ammonia extract presented herein would be notably more active. Nonetheless, it should be noted that there would be exceptions, given that Akroum and Rouibah [57] reported an MIC value of 110 µg·mL^−1^ for a cork methanolic extract against *A. alternata*. It is also worth noting that, although not comparable (given that inhibition zone values were reported instead of MIC values), other in vitro and in vivo analyses also support the antimicrobial effects of methanolic extracts of oak leaves and stems [13,58]. 

#### 3.3.2. Comparison of Efficacy with Other Natural Compounds

The results of a literature survey on the effectiveness of bioactive substances of natural origin on *M. fructigena*, *M. laxa*, and *N. parvum* fungal pathogens, on the oomycete *P. cactorum*, as well as on *E. amylovora* and *P. syringae* pv. *syringae* bacteria are compiled in Appendix A. Even if the data correspond to the same pathogens, the inhibition values listed below should be used with caution because the susceptibility profile varies depending on the isolates, the procedure for obtaining the extract, the solvent, and the testing techniques used, and also because the units used to express them differ significantly.

In the case of the *Monilinia* species, the activity of the aqueous ammonia extract of *Q. suber* cork, and that of its conjugate with COS, would be higher than those of the methanolic or n-hexane extracts of *Prunus laurocerasus* L., *Cornus mas* L., *Morus nigra* L., *Morus alba* L., and *Rosa canina* L. described by Onaran and Yanar [59], and also higher than that of the aqueous extract of *Punica granatum* L. peel [60]. It would be similar to those obtained by Mamoci et al. [61] for the n-hexane extracts of *Dittrichia viscosa* (L.) Greuter and *Ferula communis* L. It is worth noting that the lower inhibition values reported by El Khetabi et al. [62] may not be directly compared, as in that study, the essential oils were used as biofumigants. The same applies to the percentage inhibition rate reported by Andreu et al. [63], given that the units differ.

As for *N. parvum*, our research group conducted previous trials with other natural extracts against this plant pathogen using the same isolate. The efficacy of the cork extract can, thus, be directly compared to those obtained with extracts of *Equisetum arvense* L., *Urtica dioica* L., and *Silybum marianum* (L.) Gaertn. [64,65], which were lower. However, substantially higher efficacy was reported for the extract from roots of *Rubia tinctorum* L., in which inhibition was reached at 250 µg·mL^−1^ [66]. 

Concerning *P. cactorum*, the commercial CUSTOS^TM^ formulated *Allium*-based extract (MIC = 100 μg∙mL^−1^) [67], ethanolic extracts of *Pinus* spp. (MIC = 100 μg∙mL^−1^) [68], and the aqueous ammonia extract of *U. tomentosa* bark (MIC = 187.5 μg∙mL^−1^) [30] would be more effective than the *Q. suber* bark extract and its conjugate presented herein (with MIC values of 750 and 375 μg∙mL^−1^, respectively). On the other hand, commercial essential oils were minimally effective in the inhibition of this oomycete [69,70].

Regarding antibacterial activity, the inhibition results against *E. amylovora* would be comparable to those obtained by Fontana et al. [71] for the hydroethanolic, methanolic, and maltodextrin-conjugated extracts of *Moringa oleifera* Lam. leaves, with MIC values of 1000 μg∙mL^−1^; and with those obtained for hydromethanolic extracts of *P. granatum* fruits [72], of flowers or leaves of *Hibiscus syriacus* L. [73], and of flowers or leaves of *Limonium binervosum* (G.E.Sm.) C.E. Salmon [74], which were previously tested in our research group against the same isolate, with inhibition values in the 750–1500 μg∙mL^−1^ range. Finally, the lowest inhibitory levels for *P. syringae* pv. *syringae*, reported for commercial essential oils by Shabani et al. [75], cannot be directly compared, since the essential oils were utilized as biofumigants. The same holds true for the findings of Islam et al. [76], where a different methodology and different units were employed.

#### 3.3.3. Comparison of Efficacy with Conventional Fungicides

Concerning the in vitro inhibitory activity of the COS−*Q. suber* bark aqueous ammonia extract conjugate complex, it was higher than that of fosetyl-Al and much higher than that of azoxystrobin against the four pathogens, with MIC values of 1000, 750, 750, and 375 µg·mL^−1^ against *M. fructigena, M. laxa, N. parvum,* and *P. cactorum*, respectively (vs. >2000, >2000, >2000, and 2000 µg·mL^−1^ for fosetyl-Al; and >625,000, >625,000, >625,000, and ca. 6250 µg·mL^−1^ for azoxystrobin, respectively). However, in the ex situ bioassays, a ten times higher dose of the *Q. suber*-based conjugate complex (3750 µg·mL^−1^) was required to achieve full protection against *P. cactorum*. This dose represents approximately twice that of fosetyl-Al and half that of azoxystrobin. Thus, the activity of the non-optimized formulation based on the proposed natural product with a view to in-field applications may be regarded as comparable to those of these two commercial synthetic fungicides. However, future research aimed at optimizing the formulation (e.g., via combination with coadjuvants specifically designed to facilitate bark penetration) or the use of controlled release strategies (e.g., nanocarrier encapsulation [77]) may result in enhanced performance. Assessment of different treatment exposure times and application methods (e.g., spraying), which are more representative of the reality of field treatments, and inclusion of direct comparisons with conventional fungicides would also be relevant aspects to be addressed in follow-up studies.

## 4. Materials and Methods

### 4.1. Vegetal Material

The origin of the bark was the *Alcornocal de Valdegalindo* in Foncastín, Valladolid, Spain (coordinates: 41.445766, −5.014038; elevation: 707 m) (Appendix A. The bark samples (*n* = 10) were thoroughly mixed, dried, and reduced to a fine powder. According to Mota et al. [2], to prevent any damage to the trees, the corking procedure used to obtain the samples to be investigated was performed manually and during the transition from spring to summer, when their physiological circumstances were ideal for extraction. 

### 4.2. Reagents

High-molecular weight chitosan (CAS No. 9012-76-4; MW: 310,000–375,000 Da) was supplied by Hangzhou Simit Chem. & Tech. Co. (Hangzhou, China). Neutrase^TM^ 0.8 L enzyme was supplied by Novozymes A/S (Bagsværd, Denmark). Glycerin (CAS No. 56-81-5), vanillic acid 98% (4-hydroxy-3-methoxybenzoic acid, CAS No. 100-76-5), and ammonium hydroxide, 50% *v/v* aq. soln. (CAS No. 1336-21-6) were acquired from Alfa Aesar. 2-Azabicyclo[2.2.1]heptane (CAS No. 279-24-3), acetic acid (purum, 80% in H_2_O; CAS No. 64-19-7), tryptic soy agar (TSA, CAS No. 91079-40-2), and tryptic soy broth (TSB, CAS No. 8013-01-2) were supplied by Sigma–Aldrich Química (Madrid, Spain). Potato dextrose agar (PDA) was purchased from Becton Dickinson (Bergen County, NJ, USA). Alkir^®^ fungicide coadjuvant was purchased from De Sangosse Ibérica (Valencia, Spain).

Commercial fungicides used for comparison purposes, viz. Ortiva^®^ (azoxystrobin 25%; reg. no. 22000; Syngenta, Basel, Switzerland), Vondozeb^®^ (mancozeb 75%; reg. no. 18632; UPL Iberia, Barcelona, Spain), and Fosbel^®^ (fosetyl-Al 80%, reg. no. 25502; Probelte, Murcia, Spain) were kindly provided by the Plant Health and Certification Service (CSCV) of Gobierno de Aragón.

### 4.3. Phytopathogen Isolates

*M. laxa* (MYC-1580) and *M. fructigena* (MYC-1579) were supplied by the Mycology lab at the Center for Research and Agrifood Technology of Aragón (CITA, Zaragoza, Spain) as subcultures on PDA. *P. cactorum* (CRD Prosp/59) and the bacterial isolate *P. syringae* pv. *syringae* (CRD 17/105) were supplied by the Regional Diagnostic Center of Aldearrubia (Junta de Castilla y León, Spain) as subcultures in PDA or TSA, respectively. The *N. parvum* isolate (code ITACYL_F111, isolate Y-091-03-01c) was supplied in a lyophilized vial (later reconstituted and refreshed as a subculture on PDA) by the Instituto Tecnológico Agrario de Castilla y León (ITACYL; Valladolid, Spain). *E. amylovora* (NCPPB 595) was obtained from the Spanish Type Culture Collection (CECT; Valencia, Spain).

### 4.4. Preparation of Bark Extracts, Chitosan Oligomers, and Conjugate Complexes

The preparation of the *Q. suber* bark extract followed the method previously described in [30]; briefly, the bark powder sample was first digested in an aqueous ammonia solution for 2 h, then sonicated in pulsed mode (with a 2 min stop every 2.5 min) for 10 min using a probe-type ultrasonicator (model UIP1000hdT; 1000 W, 20 kHz; Hielscher Ultrasonics, Teltow, Germany), and then allowed to stand for 24 h. Finally, the solution was centrifuged at 9000 rpm for 15 min, and the supernatant was filtered through Whatman No. 1 paper.

Chitosan oligomers (COS) were prepared according to the procedure described in the work by Santos-Moriano et al. [78], with the modifications indicated in [77]. In brief, commercial chitosan (MW = 310–375 kDa) was dissolved in aqueous 1% (*w/w*) acetic acid, and, after filtration, the filtrate was neutralized with aqueous 4% (*w/w*) NaOH. The precipitate was collected and washed thoroughly with hot distilled water, ethanol, and acetone. Purified chitosan was obtained by drying. The degree of deacetylation (DD) was determined to be 90% according to Sannan et al. [79]. A total of 20 g of purified chitosan was dissolved in 1000 mL of Milli-Q water by adding 20 g of citric acid under constant stirring at 60 °C. Once dissolved, the commercial proteolytic preparation Neutrase^TM^ 0.8 L (a protease from *Bacillus amyloliquefaciens* Priest, Goodfellow, Shute & Berkeley) was added to obtain a product enriched in deacetylated chitooligosaccharides and to degrade the polymer chains. The mixture was sonicated for 3 min in 1 min of sonication/1 min without sonication cycles to keep the temperature in the 30–60 °C range. The molar mass of the COS samples was determined by measuring the viscosity, in agreement with the method outlined by Yang et al. [80], in a solvent of 0.20 mol·L^−1^ NaCl + 0.1 mol·L^−1^ CH_3_COOH at 25 °C using an Ubbelohde capillary viscometer. The molar mass was determined using the Mark–Houwink equation [η] = 1.81 × 10^−3^ M^0.93^ [81]. At the end of the process, a solution with a pH in the four to six interval with oligomers of molecular weight <2 kDa was obtained, with a polydispersity index of 1.6, within the usual range reported in the literature [82].

The COS–bark extract and COS−main bioactive compounds conjugate complexes were obtained by mixing the respective solutions in a 1:1 (*v/v*) ratio, followed by sonication for 15 min in 5 3-min pulses (so that the temperature did not exceed 60 °C). Attenuated total-reflectance Fourier transform infrared (ATR-FTIR) spectroscopy was used to confirm the formation of the conjugate complexes.

### 4.5. Extract and Conjugate Complexe Characterization

The aqueous ammonia *Q. suber* bark extract was studied by gas chromatography–mass spectrometry (GC-MS) at the Research Support Services (STI) at Universidad de Alicante (Alicante, Spain), using a gas chromatograph model 7890A coupled to a quadrupole mass spectrometer model 5975C (both from Agilent Technologies, Santa Clara, CA, USA). Chromatographic conditions were as follows: 3 injections/vial, injection volume = 1 µL; injector temperature = 280 °C, in splitless mode; initial oven temperature = 60 °C, after 2 min, followed by an increase of 10 °C/min up to a final temperature of 300 °C, after 15 min. The chromatographic column used for the separation of the compounds was an Agilent Technologies HP-5MS UI of 30 m in length, 0.250 mm in diameter, and with 0.25 µm film. The conditions of the mass spectrometer were as follows: temperature of the electron impact source of the mass spectrometer = 230 °C and of the quadrupole = 150 °C; ionization energy = 70 eV. Test mixture 2 for apolar capillary columns according to Grob (Supelco 86501) and PFTBA tuning standards were used for equipment calibration. The identification of the components was based on a comparison of their mass spectra and retention time with those of the authentic compounds and by computer matching with the database of the National Institute of Standards and Technology (NIST11) and Adams [83].

### 4.6. In Vitro Antimicrobial Activity Assessment

The antifungal/anti-oomycete activity of the different treatments (including *Q. suber* bark extract, some of its main constituents, the conjugates of all of them, and certain commercial synthetic fungicides) was determined by the agar dilution method according to the EUCAST antifungal susceptibility testing standard procedures [84], incorporating stock solution aliquots into the pouring PDA medium to provide final concentrations in the 62.5–1500 μg·mL^−1^ range. Mycelial plugs (⌀ = 5 mm), from the margin of 1-week-old PDA cultures of *M. laxa*, *M. fructigena*, and *N. parvum*, and 2-week-old cultures of *P. cactorum*, were transferred to plates, incorporating the above treatment concentrations (3 plates/concentration, with 2 replicates). *Neofusicoccum parvum* and *P. cactorum* plates were incubated at 25 °C in the dark for 1 and 2 weeks, respectively. In the case of *M. laxa* and *M. fructigena*, incubation was carried out at 22 °C in the dark for 1 week. PDA medium without any modification was used as the control. Mycelial growth inhibition was estimated according to the formula ((*d_c_ − d_t_*)/*d_c_*) × 100, where *d_c_* and *d_t_* represent the mean diameters of the control and treated fungal colonies, respectively. The effective concentrations (EC_50_ and EC_90_) were estimated using PROBIT analysis in IBM SPSS Statistics v.25 software (IBM; Armonk, NY, USA). The level of interaction, i.e., the synergy factor, was estimated according to Wadley’s method [85].

The antibacterial activity was assessed according to the CLSI standard M07–11 [86], using the agar dilution method to determine the minimum inhibitory concentrations (MIC). In short, an isolated colony of *E. amylovora* was incubated at 30 °C for 18 h in a TSB liquid medium. Serial dilutions were then performed, starting from a concentration of 10^8^ CFU·mL^−1^, to obtain a final inoculum of ~10^4^ CFU·mL^−1^. The bacterial suspensions were then delivered to the surface of the TSA plates, to which the treatments had previously been added at concentrations ranging from 62.5 to 1500 μg·mL^−1^. Plates were incubated at 30 °C for 24 h, and readings were taken after 24 h. In the case of *P. syringae* pv. *syringae*, the same procedure was followed, although it was grown at 25 °C for 48 h. MICs were determined in the agar dilutions as the lowest concentrations of the bioactive products at which no bacterial growth was visible. All experiments were run in triplicate, with 3 plates per treatment/concentration.

### 4.7. Protection Tests on Artificially Inoculated Excised Stems

The efficacy of the treatment was tested by artificial inoculation of excised stems in controlled laboratory conditions. Inoculation was performed according to the procedure proposed by Matheron [87], with the modifications described by Álvarez Bernaola [88]. Briefly, using a grafting knife, young stems of healthy ‘Garnem’ (*Prunus amygdalus* × *P. persica*) rootstock with a 1.5 cm diameter were cut into 10 cm long sections. The excised stem pieces were immediately wrapped in moistened sterile absorbent paper, while the produced wounds were painted with Mastix^®^.

In the laboratory, the freshly excised stem segments were first immersed in a NaClO 3% solution for 10 min, immersed in ethanol 70% for 10 min, and then thoroughly rinsed four times with bidistilled sterile water, to avoid superficial contaminants in the tissue. Some of the stem segments (*n* = 10) were soaked for 1 h in distilled water as a control, while the remaining stem segments were soaked for 1 h in aqueous solutions to which an appropriate amount of the conjugate complex of COS–*Q. suber* bark extract had been added to obtain MIC, MIC × 5, and MIC × 10 concentrations (*n* = 10 segments/concentration). Alkir^®^ coadjuvant (1% *v/v*) was added to all the solutions (including the control) to facilitate the moistening and penetration of the treatment into the bark.

The stem pieces were allowed to dry, and the bark was carefully removed with a scalpel to reveal the cambium. The bark was then placed on an agar Petri dish, and subsequently inoculated by placing a plug (⌀ = 5 mm), from the margin of 2-week-old PDA cultures of *P. cactorum,* on the center of the inner surface of the bark. After inoculation, stem segments were incubated in a humid chamber for 3–4 days at 24 °C, 95–98% RH.

The efficacy of the treatments was evaluated by measuring the lengths of the cankers that developed at the inoculation sites. Finally, the *P. cactorum* strain was re-isolated and morphologically identified from the lesions to fulfill Koch’s postulates.

### 4.8. Statistical Analyses

The results of the in vitro mycelium growth inhibition experiments were statistically analyzed using one-way analysis of variance (ANOVA), followed by a post hoc comparison of means using the Tukey test at *p* < 0.05, given that the homogeneity and homoscedasticity requirements were met, according to the Shapiro–Wilk and Levene tests. For the ex situ assays on excised ‘Garnem’ stems, in which normality and homoscedasticity requirements were not met, the Kruskal–Wallis nonparametric test was used instead, with the Conover–Iman test for post hoc multiple pairwise comparisons. R v.4.2.2 statistical software was used for all the statistical analyses [89].

## 5. Conclusions

In vitro antifungal/anti-oomycete and antibacterial tests showed the moderate–high activity of the aqueous ammonia extract of *Q. suber* cork, with EC_90_/MIC values ranging from 675 to 3450 μg·mL^−1^, depending on the pathogen tested. Upon conjugation with chitosan oligomers, substantial antimicrobial activity enhancement was observed, resulting in inhibitory values in the 340–801 μg·mL^−1^ range. The observed in vitro activity was lower than that of mancozeb, but remarkably higher than those of azoxystrobin and fosetyl-Al commercial fungicides, which were taken as a reference, and is among the highest reported for other natural compounds. In a first approximation, glycerin and vanillic acid, identified by gas chromatography–mass spectroscopy and tested as pure compounds, would be responsible for the observed activity. To assess the applicability of the treatment in more realistic conditions, the COS–*Q. suber* bark extract conjugate complex was further tested ex situ for the protection of excised stems artificially inoculated with *P. cactorum*, observing that a 10 times higher dose than the MIC determined in vitro (i.e., 3750 μg·mL^−1^) was required to achieve high protection, which would be twice the dose recommended for fosetyl-Al (2000 μg·mL^−1^) and approximately half that of azoxystrobin (ca. 6250 μg·mL^−1^). In view of these promising results, a possible valorization pathway for leftovers from the bottle stopper industry (cork fragments, granulates, and dust) as a source of phytochemicals for crop protection can be put forward.

## Figures and Tables

**Figure 1 plants-11-03415-f001:**
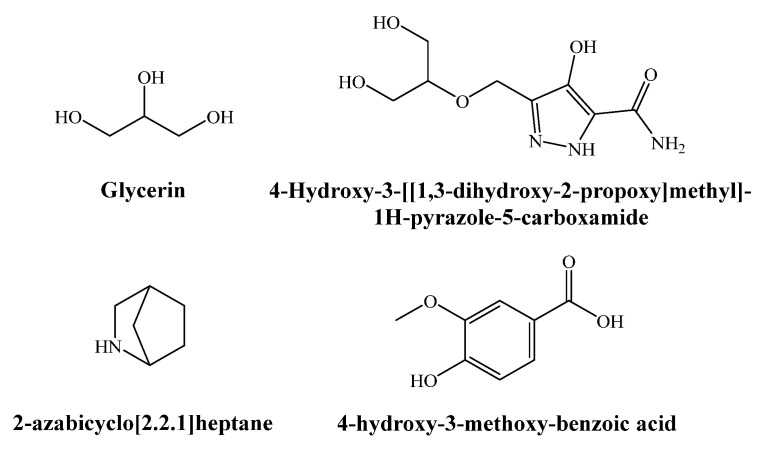
Main phytochemicals identified in the aqueous ammonia extract of *Quercus suber* bark.

**Figure 2 plants-11-03415-f002:**
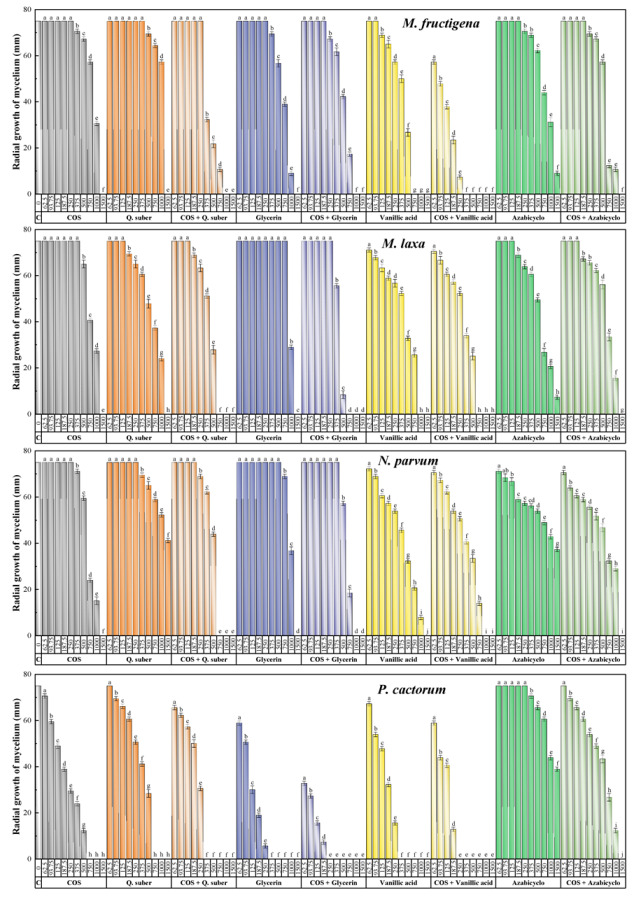
Inhibition of the radial growth of the mycelium in in vitro tests performed in PDA medium with different concentrations (in the 62.5–1500 µg·mL^−1^ range) of chitosan oligomers (COS), *Q. suber* bark extract, three of its main phytochemical constituents, and their respective conjugated complexes. The same letters for the above concentrations mean that they are not significantly different at *p* < 0.05. Error bars represent standard deviations. Azabicyclo = 2-azabicyclo[2.2.1]heptane.

**Figure 3 plants-11-03415-f003:**
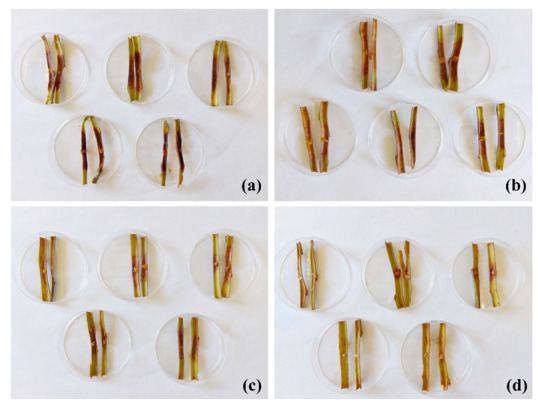
Canker lengths observed in ‘Garnem’ excised stems artificially inoculated with *P. cactorum* and treated with the COS–*Q. suber* bark extract conjugate complex at different concentrations: (**a**) control, no treatment; (**b**) MIC = 375 μg·mL^−1^; (**c**) MIC × 5 = 1875 μg·mL^−1^; (**d**) MIC × 10 = 3750 μg·mL^−1^. Only half of the replicates per treatment are shown.

**Table 1 plants-11-03415-t001:** Effective concentrations (expressed in µg·mL^−1^) against *M. fructigena*, *M. laxa*, *N. parvum,* and *P. cactorum* of chitosan oligomers (COS), the aqueous ammonia extract of *Q. suber* bark, and three of its main constituents, alone and upon conjugation with COS.

Treatment	Effective Concentration	*M. fructigena*	*M. laxa*	*N. parvum*	*P. cactorum*
COS	EC_50_	940.9	808.0	677.5	200.8
	EC_90_	1356.0	1342.7	1198.2	592.8
*Q. suber* bark	EC_50_	1073.6	684.4	1738.7	409.4
	EC_90_	1466.8	1360.4	3449.4	674.9
COS–*Q. suber* bark	EC_50_	314.2	446.4	508.0	224.4
	EC_90_	801.1	671.8	706.3	339.7
Glycerin	EC_50_	773.5	1012.0	964.3	115.5
	EC_90_	1216.1	1343.8	1470.8	267.1
COS–Glycerin	EC_50_	545.7	420.5	622.2	54.5
	EC_90_	880.6	623.2	890.7	185.1
Vanillic acid	EC_50_	440.2	498.8	448.0	159.3
	EC_90_	668.8	931.0	1108.7	313.1
COS–Vanillic acid	EC_50_	128.5	351.3	424.0	129.2
	EC_90_	256.1	669.9	861.6	214.5
Azabicyclo	EC_50_	862.0	636.0	1444.9	1677.1
	EC_90_	1556.3	1420.7	8779.8	6805.6
COS–Azabicyclo	EC_50_	606.5	696.8	667.4	567.7
	EC_90_	1117.1	1244.0	1399.9	1198.9

Azabicyclo = 2-azabicyclo[2.2.1]heptane.

**Table 2 plants-11-03415-t002:** Synergy factors for the conjugate complexes estimated according to Wadley’s method.

Treatment	Effective Concentration	*M. fructigena*	*M. laxa*	*N. parvum*	*P. cactorum*
COS–*Q. suber* bark	EC_50_	3.19	1.66	1.92	1.20
	EC_90_	1.76	2.01	2.52	1.86
COS–Glycerin	EC_50_	1.56	2.14	1.28	2.69
	EC_90_	1.46	2.16	1.48	1.99
COS–Vanillic acid	EC_50_	6.08	1.76	1.27	1.38
	EC_90_	2.60	1.64	1.34	1.91
COS–Azabicyclo	EC_50_	1.48	1.02	1.38	0.63
	EC_90_	1.30	1.11	1.51	0.91

Azabicyclo = 2-azabicyclo[2.2.1]heptane.

**Table 3 plants-11-03415-t003:** Radial growth of mycelium of *M. laxa*, *M. fructigena*, *N. parvum,* and *P. cactorum* in in vitro assays performed on a PDA medium with different concentrations (the recommended dose, a tenth of the recommended dose, and ten times the recommended dose) of three commercial synthetic fungicides.

Commercial Fungicide	Pathogen	Radial Growth of Mycelium (mm)			Inhibition (%)		
		Rd/10	Rd *	Rd × 10	Rd/10	Rd *	Rd × 10
Azoxystrobin	*M. laxa*	33.7	30.3	29.2	55	59.6	61.1
	*M. fructigena*	60.3	56	33.2	19.6	25.3	55.8
	*N. parvum*	74.4	72.5	68.1	0.9	3.4	9.2
	*P. cactorum*	6	0	0	92	100	100
Mancozeb	*M. laxa*	0	0	0	100	100	100
	*M. fructigena*	0	0	0	100	100	100
	*N. parvum*	0	0	0	100	100	100
	*P. cactorum*	0	0	0	100	100	100
Fosetyl-Al	*M. laxa*	72.1	13.3	0	3.9	82.2	100
	*M. fructigena*	82.4	18.4	0	0	75.5	100
	*N. parvum*	59.1	8.7	0	21.2	88.4	100
	*P. cactorum*	64	0	0	14.7	100	100

* Rd stands for recommended dose, i.e., 62.5 mg·mL^−1^ of azoxystrobin (250 g·L^−1^ for Ortiva^®^, azoxystrobin 25%), 1.5 mg·mL^−1^ of mancozeb (2 g·L^−1^ for Vondozeb^®^, mancozeb 75%) and 2 mg·mL^−1^ of fosetyl-Al (2.5 g·L^−1^ for Fosbel^®^, fosetyl-Al 80%). The radial growth of the mycelium for the control (PDA) was 75 mm. All mycelial growth values (in mm) are average values (*n* = 2).

**Table 4 plants-11-03415-t004:** Antibacterial activity against *E. amylovora* and *P. syringae* pv. *syringae* of chitosan oligomers (COS), the aqueous ammonia extract of *Q. suber* bark, and its main constituents, alone and upon conjugation with COS. Positive and negative signs indicate the presence/absence of bacterial growth.

Pathogen	Compound	Concentration (μg·mL^−1^)									
		62.5	93.75	125	187.5	250	375	500	750	1000	1500
*E. amylovora*	COS	+	+	+	+	+	+	+	+	+	−
	*Q. suber* bark	+	+	+	+	+	+	+	+	−	−
	Glycerin	+	+	+	+	+	+	−	−	−	−
	Vanillic acid	+	+	+	+	+	+	+	−	−	−
	Azabicyclo	+	+	+	+	+	+	+	+	+	+
	COS−*Q. suber* bark	+	+	+	+	+	+	+	−	−	−
	COS−Glycerin	+	+	+	+	+	−	−	−	−	−
	COS−Vanillic acid	+	+	+	+	+	+	−	−	−	−
	COS−Azabicyclo	+	+	+	+	+	+	+	−	−	−
*P. syringae* pv. *syringae*	COS	+	+	+	+	+	+	+	+	−	−
	*Q. suber* bark	+	+	+	+	+	+	+	−	−	−
	Glycerin	+	+	+	+	+	+	+	−	−	−
	Vanillic acid	+	+	+	+	+	+	+	−	−	−
	Azabicyclo	+	+	+	+	+	+	+	+	+	+
	COS−*Q. suber* bark	+	+	+	+	+	−	−	−	−	−
	COS−Glycerin	+	+	+	+	+	−	−	−	−	−
	COS−Vanillic acid	+	+	+	+	+	+	−	−	−	−
	COS−Azabicyclo	+	+	+	+	+	+	+	−	−	−

Azabicyclo = 2-azabicyclo[2.2.1]heptane.

**Table 5 plants-11-03415-t005:** Results of the Kruskal–Wallis test, followed by multiple pairwise comparisons using the Conover–Iman procedure for mean lengths of cankers in excised stems after inoculation with *P. cactorum*. The mean rank values accompanied by the same letters are not significantly different (*p*-value (one-tailed) < 0.0001, α = 0.05).

Sample	Mean of Ranks	Groups		
MIC × 10	5.6	A		
MIC × 5	18.1		B	
MIC	28.9			C
Non-treated control	29.4			C

## Data Availability

The data presented in this study are available upon request from the corresponding author. The data are not publicly available due to their relevance to an ongoing Ph.D. thesis.

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
