# Peer review of "Valorization of Quercus suber L. Bark as a Source of Phytochemicals with Antimicrobial Activity against Apple Tree Diseases"

_plants, 2022, doi:10.3390/plants11243415_

Round 1
Reviewer 1 Report
The present study is an interesting research about the valorisation of the residue of Quercus suber L. bark as a source of phytochemical compounds with antimicrobial activity against tree diseases.
Even it is a complete study I have some comments or issues to be fixed.
Abstract is complete, giving a complete idea about what the study is about
The introduction is too long and should be more focused in the state of the art of the actual problems and possible uses of cork extracts. In the present form introduction goes a bit far from the topic in some lines, including references and examples that are not needed since the text and information is already large enough. Introduction must be more concise.
An important point is the English review. In other parts of the manuscript the English is of a better quality but the introduction needs much more review by an English native speaker. As an example of English review needs is the sentence in lines 78 and 79.
Material and Methods:
Material and methods are in general well described and also the English quality is better, probe of the authors experience in this field.
In the point 4.7 it is described the inoculation of excised stems. It is not described the NaClO concentration (%) and also not described how are the stems rinsed. Several protocols talk about the times the plant material must be rinsed in order to avoid residues of NaClO. At this point I would say that I miss the introduction of a positive control like the use of a fungicide in the inoculation to compare directly. For me 1 hour inoculation is a long time that does not show the reality of a field treatment even if a protocol is described. May be it is something to look for future studies.
Results
Figures are of a high value but in some cases I miss some information
Results description is also good and of a good extension. Minor English review is needed in this part only.
Figure 2 is quite large, so there is a good chance of including name of fungi tested inside and the figure and not all the information at the figure description.
Discussion
The discussion is large and complex, since it is difficult to find comparable results, and in many cases is more a well-formed supposition. In general the discussion goes in the direction that no many comparisons can be made since there are not many results based on same methodology.
This is difficult also because different growing conditions result in different profiles or at least different compound concentrations
In this case, the direct comparison with conventional fungicides or same extraction methods are the most informative comparatives.
As stabilised, lack of formulation of actual extracts is one of the problems as regards the effectivity of the treatment in ex situ assays
Reviewer 2 Report
Sánchez‐Hernández et al. presented an interesting study on how cork aqueous ammonia extracts inhibit the growth of pathogenic fungi and bacteria. The authors first analyzed the spectrum of chemical compounds in the extracts by using gas chromatography‐mass spectrometry and identified four main chemicals which were then tested either alone or coupled with chitosan oligomers (COS). The cork extract, COS, and the combination of both were examined as well. Importantly, cork extract mixed with chitosan oligomers showed a substantial synergistic effect thus providing antimicrobial activity comparable to or exceeding conventional synthetic fungicides. The authors conclude with a proposal for the possible application of industrial waste from the bottle stopper industry. The research is based on multiple experimental approaches (i.e., measurement of radial growth inhibition of oomycetes, antibacterial activity, and canker lengths in excised stems). The statistical analysis is rigorous and thorough. Finally, the manuscript is well-written and organized.
I have only several minor comments and questions to be addressed before recommending the article for acceptance:
1) Four major compounds identified by GC-MS were glycerin, 4‐ hydroxy‐3‐[[1,3‐dihydroxy‐2‐propoxy]methyl]‐1H‐pyrazole‐5‐carboxamide, 2-azabicyclo[2.2.1] heptane, and vanillic acid. However, the authors tested only three of them dropping the 4‐ hydroxy‐3‐[[1,3‐dihydroxy‐2‐propoxy]methyl]‐1H‐pyrazole‐5‐carboxamide. What was the rationale for that?
2) Figure 1 presents the main phytochemicals in the crock extract and contains nonanedioic acid, however, there is no mention of this compound in the text above the figure (it is presented only in the Discussion). Moreover, as can be inferred from table S1, nonanedioic is not one of the major compounds reaching only 0.4%. It is thus unclear, why this compound was included in Figure 1. Please, explain.
3) What approach was used to determine anthropogenic contaminants?
4) In the 2.2.2 header “In vitro” should be italicized;
5) It would be informative to expand table S2 with the antimicrobial activity of glycerin as presented for vanillic acid and azabicyclo derivates;
6) Please delete the part of the article template (lines 406-408);
7) Please italicize “post hoc” (line 555);
8) Please provide the version of the R language used.
